# Serial, Visually-Evoked Potentials for the Assessment of Visual Function in Patients with Craniosynostosis

**DOI:** 10.3390/jcm8101555

**Published:** 2019-09-27

**Authors:** Mostafa M. Haredy, Alki Liasis, Amani Davis, Kathleen Koesarie, Valeria Fu, Joseph E. Losee, Jesse A. Goldstein, Ken K. Nischal

**Affiliations:** 1Department of Plastic Surgery—Cleft-Craniofacial Center, Children’s Hospital of Pittsburgh, University of Pittsburgh Medical Center, Pittsburgh, PA 15201, USA; mostafaharedy84@gmail.com (M.M.H.); joseph.losee@chp.edu (J.E.L.); jesse.goldstein@chp.edu (J.A.G.); 2Plastic Surgery Department—Cleft and Craniofacial Unit, Sohag University Hospital, Sohag 82511, Egypt; 3Department of Ophthalmology, Children’s Hospital of Pittsburgh, University of Pittsburgh Medical Center, UPMC Eye Center, Pittsburgh, PA 15201, USA; liasisa@chp.edu (A.L.); amanidavis@wustl.edu (A.D.); valeria_fu@yahoo.com (V.F.); 4School of Medicine, University of Pittsburgh, Pittsburgh, PA 15213, USA; Koesarie.Kathleen@medstudent.pitt.edu

**Keywords:** craniosynostosis, visual function, visually-evoked potentials

## Abstract

This study aimed to evaluate the effect of craniofacial surgical intervention on the visual pathway’s function by comparing pre- to post-operative patterned, visually-evoked potentials (pVEP). A retrospective review was conducted on craniosynostosis patients who had pre- and post-craniofacial surgery pVEP testing. The pVEP measured grade in terms of amplitude latency and morphology of the waveforms. The pre- and post-operative results were compared. The study identified 63 patients (mean age at preoperative pVEP of 16.9 months). Preoperatively, 33 patients (52.4%) had abnormal pVEP. Nine patients had evidence of intracranial hypertension, and of those, eight (88.9%) had abnormal pVEP. Within 6 months postoperatively, 24 of 33 patients (72.7%) with abnormal preoperative pVEP developed normal postoperative pVEP, while all 30 patients with normal preoperative VEP maintained their normal results postoperatively. Significant improvements in pVEP latency in patients with broad or delayed latency waveforms was evident for subjects with preoperative grades 2–4 (grade 2, *p* = 0.015; grade 3, *p* = 0.029; grade 4; *p* = 0.007), while significant postoperative increase in amplitude was significant for patients with abnormally low amplitude grade 3 and 5 waveforms (grade 3, *p* = 0.011; grade 5, *p* = 0.029). Serial pVEP testing represents a useful tool for the early detection of visual pathway dysfunction and follow up visual pathway function in craniosynostosis. Surgical intervention for craniosynostosis can result in the reversal of preoperative pVEP abnormalities seen in these patients, resulting in the normalization of the pVEP waveform, amplitude and latency, depending on the preoperative pVEP abnormality.

## 1. Introduction

Craniosynostosis is the premature closure of one or more of the cranial sutures, with a birth prevalence of 3.5 to 4.5 per 10,000 live births [1]. Children with craniosynostosis may suffer from functional sequalae, including developmental delay, behavioral disturbances, learning disabilities, and visual loss [2]. The latter may be due to optic neuropathy, amblyopia, corneal exposure, or any combination of these [3]. In craniosynostosis, optic neuropathy may occur due to raised intracranial pressure (ICP), because of craniocerebral disproportion [4] or hydrocephalus [5], or be due to cerebral hypoperfusion, especially in cases of syndromic craniosynostosis associated with obstructive sleep apnea (OSA) [6] and intracranial venous anomalies [7]

Optic neuropathy can occur without papilledema in cases of craniosynostosis, and although papilledema has high specificity and is widely used as a screening tool for detection of raised ICP, it has low sensitivity [8,9]. The early detection of optic neuropathy is of utmost importance in patients with craniosynostosis to prevent permanent vision loss that is reported to occur in syndromic craniosynostosis, even after vault expansion surgery [2,10]. 

Visually-evoked potentials (VEP) are electrical signals generated in the occipital lobe in response to visual stimuli and reflect the integrity of the visual pathway from the optic nerve to the visual cortex. Pattern reversal visually-evoked potentials (pVEP) have been used to detect impairments in the visual pathway functioning of patients with craniosynostosis, and they have shown to be a useful non-invasive tool to detect early visual pathway dysfunction in these patients [11]. The aim of this study was to detect the prevalence of abnormal pVEP in a cohort of patients with craniosynostosis and to evaluate the effect of craniofacial surgical intervention on the visual pathway function by comparing pre- to post-operative pVEP.

## 2. Materials and Methods

### 2.1. Study Population

This study was conducted in accordance with the Declaration of Helsinki, and the protocol was approved by the institutional review board of University of Pittsburgh Medical Center (IRB number PRO16090260). The study involved a retrospective review of patients with craniosynostosis presenting in the Cleft-Craniofacial Center of Children’s Hospital of Pittsburgh, between January 2012 and October 2017, who had pre- and post-operative reversal pVEP testing. Patients with bilateral amblyopia were excluded from our study, as amblyopia is known to affect VEP measures [12], and the effects of unilateral amblyopia were overcome in our study, as only results of binocular VEP findings were included in this analysis.

### 2.2. pVEP Recording

The recording methods were previously published [13]. Briefly, patients sat on a comfortable chair one meter from a plasma monitor, viewing, with both eyes open, a mean screen luminance of 95 cd/m^2^. pVEP were evoked by a black and white 3 Hz reversing checkerboard presented in a 35-degree field. Individual check elements used had a side subtense ranging from 100 to 12.5 arc. Patients were asked to attend a fixation spot (0.25 degrees) in the center of the screen. Closed circuit T.V. was used to monitor fixation and data collection was interrupted during any fixation instability. In younger children, an additional vision scientist was also present during testing to orient the child to fixation point. The ongoing EEG was also monitored for slow-wave activity associated with drowsiness. Silver–silver chloride electrodes placed at three electrode site over the occipital region (Oz, O1, and O2) referred to a mid-frontal electrode (Fz) with a cephalic ground electrode. The impedances of all electrodes were balanced and maintained below 5 kΩ throughout the recordings. The EEG was digitized using a sampling rate of 1 kHz and a band-pass filter of 0.312–100 Hz. The amplifiers had a fixed gain of 8with an input range of ±0.5 V (Espion E 3.0 by Diagnosys, 83 Cambridge, UK). If the presence of line noise was present, despite balanced electrode impedances, the inbuilt line filter was applied during the recordings. Epochs of 300 ms (−15 to 285 ms) were averaged online, with any epochs exceeding ±200 μv automatically rejected. A minimum of two reproducible trials were recorded for each stimulus. Reproducibility was determined using the criteria obtained from a previous study [14]. The two trials were deemed reproducible if the amplitude did not vary more than 15% and latency by 2%. In all cases, the grand average of the acquired trials was subject to analysis. Measurements were taken from those evoked by the smallest test check for each individual subject, which was considered the individual’s threshold stimulus and was used for postoperative follow up of the visual function. This was chosen by the test check size that yielded a reproducible response above the level of noise defined by the characteristics of a reproducible response described above. Table 1 shows the check sizes used for each pVEP grade of our patients. The threshold pVEP measures have shown to be a more sensitive index of optic nerve compromise than larger stimuli [15].

### 2.3. pVEP Grading

The P100 measures of pVEP were compared with age matched control data of the pediatric ophthalmic laboratory. As shown in Table 1, the results were then graded according to a previously published classification by Thompson et al. [11]. The grading of the VEP responses was carried out by 2 senior electrophysiologists. Under this classification, grade 1 represents normal pVEP; grade 2 represents the mildest pVEP affection with waveform broadening (>70 ms, measuring the latency between the N80 and the N135); grade 3 includes patients with broad waves and reduced amplitude; grade 4 is for more severe abnormalities, consisting of prolonged latencies; and grade 5 represents the highest level of pVEP abnormality, with reduced amplitude and prolonged latency [11].

### 2.4. Statistical Analysis

Statistical analysis was performed using IBM SPSS Statistics (Version 24.0., IBM Corp., Armonk, NY, USA) Statistical significance was defined as *p* < 0.05. The relationship between age and preoperative VEP grade was evaluated using the Kruskal–Wallis test. The Chi-Square test was used to test the relationship between syndromic association and abnormal preoperative pVEP. The mean pre- and post-operative pVEP amplitude and latency for the whole cohort and for individual grades were calculated and compared using paired *t*-tests. 

## 3. Results

### 3.1. Study Population

The study identified 67 craniosynostosis patients who had pre- and post-operative pVEP during the study period. Four patients were excluded; two had bilateral amblyopia, one had optic nerve hypoplasia, and the fourth patient had inconsistent postoperative results, as the patient was inattentive during the test. Sixty-three patients were included in this study (mean age at preoperative pVEP 16.9 months; range 2 months–10 years). Twenty-two patients (34.9%) had syndromic/complex craniosynostosis, while 41 patients (65.1%) had nonsyndromic, single suture synostosis. The mean interval between preoperative pVEP testing and surgery was 2.8 months. Postoperative pVEP testing was performed for all patients within 6 months following surgery, with a mean of 3.9 months. A total of 28 (44.4%) of our patients had an additional pVEP test more than 6 months after surgery (mean 11.5 ± 5.6 months). Only the results from postoperative pVEP tests done within 6 months were included in the statistical analyses of postoperative pVEP amplitude and latency changes.

### 3.2. Preoperative VEP Grading Results

Thirty patients (47.6%) presented with grade 1 (normal) pVEP, while 33 patients (52.4%) had grade 2 or higher (abnormal) pVEP. Of the patients with abnormal pVEP, 10 patients (30.3%) had grade 2, nine patients (27.3%) had grade 3, 10 patients (30.3%) had grade 4, and four patients (12.1%) had grade 5 (Table 1). There was no significant relationship between age at preoperative pVEP and pVEP grade (*p* = 0.51). Of the 22 patients with syndromic/complex craniosynostosis, 14 (63.6%) had abnormal pVEP, while 19 of the 41 patients (46.3%) with single suture disease demonstrated abnormal pVEP. This increased association between syndromic/complex craniosynostosis and abnormal pVEP, however, was not statistically significant (*p* = 0.19).

### 3.3. Intracranial Hypertension

A total of nine patients showed evidence of intracranial hypertension in this cohort; of these, five had raised ICP, seen in ICP monitoring, while fundus examination revealed papilledema in the remaining four patients (with no ICP monitoring performed). Four of the five patients who had raised ICPs upon ICP monitoring showed abnormal pVEP, while all four patients with papilledema upon fundus examination performed abnormally in the pVEP test.

### 3.4. Surgical Intervention

The craniofacial surgical intervention performed depended on the patient age, craniosynostosis type, and associated skull deformity. Forty-nine patients (77.8%) underwent vault remodeling surgery (front-orbital advancement in 27 patients, anterior or posterior vault remodeling in 14 patients, and total/pi procedure in eight patients), six patients had posterior vault distraction osteogenesis, five patients had springs cranioplasty, and strip craniectomy was performed in three patients.

### 3.5. Postoperative VEP Changes

Within 6 months, a total of 54 patients (85.7%) maintained or established normal pVEP postoperatively, compared to the 47.6% who had a normal preoperative pVEP test. The postoperative pVEP showed a statistically significant decrease (improvement) in latency (*p* < 0.001), and an increase (improvement) in amplitude (*p* = 0.02) compared to preoperative measures for the total study cohort. 

On stratifying postoperative pVEP by preoperative pVEP grades, all patients with preoperative grade 1 maintained their normal pVEP (Figure 1). As expected, there was no statistical change in the latency (*p* = 0.18) or amplitude (*p* = 0.24), as the response was and remained within normal limits. For patients with grade 2, all 10 cases moved to grade 1 postoperatively; i.e., normalized their pVEP waveform breadth. The change from grade 2 to grade 1 also resulted in significant decrease in the latency (*p* = 0.02), with no significant changes in amplitude (*p* = 0.11) (Table 2 and Table 3). Sample waveforms are shown in Figure 2.

Eight of nine (88.9%) patients with grade 3 moved to grade 1 postoperatively. There was a significant postoperative increase in amplitude (*p* = 0.01) and a reduction in latency (*p* = 0.03). For grade 4, five of 10 (50%) patients normalized their pVEP grade latency postoperatively, while one patient moved to grade 2 (normalization of latency, but still broad waveform). The remaining four patients showed persistent, prolonged, although improved, latency. A significant decrease in latency for patients with grade 4 (*p* = 0.007) was found postoperatively, with no significant change in amplitude (*p* = 0.37). The five patients with a persistent, postoperatively abnormal pVEP had subsequent pVEP done within 1 year later, and all but one of them developed normal pVEP.

This gradual normalization of latency was also appreciated in patients with grade 5, where only one patient had normal latency within 6 months after surgery (*p* = 0.07), but the other three patients normalized their latency over a period of 18 months postoperatively. On the other hand, the amplitude of all four patients normalized within 6 months postoperatively (*p* = 0.03) (Figure 3).

While 24 of the 33 patients (72.7%) with abnormal preoperative pVEP developed normal postoperative results within 6 months postoperatively, a total of 31 (of the 33 patients; 93.9%) developed normal postoperative pVEP over a period of 18 months postoperatively.

## 4. Discussion

Children with craniosynostosis may suffer from visual loss which can result from optic neuropathy, amblyopia, or corneal exposure [16]. In craniosynostosis, the incidence of irreversible visual failure due to intracranial hypertension has been reported to be 3% [17], and several studies have reported the presence of intracranial hypertension without evidence of papilledema [9,18]. Behavioral visual acuity tests in infants and young children may be difficult to elicit and can be misleading, as they improve with the age and cooperation of the child, such that permanent damage can occur before we can obtain reliable and reproducible measurements of visual acuity [19].

The first report of use of VEP in craniosynostosis was introduced by Gupta et al., who noticed the presence of abnormal flash VEP (fVEP) in 11 of 66 eyes [20]. Similarly, Mursch et al., using fVEP in 48 patients and pVEP in four patients with craniosynostosis, reported pathological VEP in 14 of 52 patients [21]. 

It is well established that pVEP is less variable across subjects and more sensitive to detect early visual pathway dysfunction than VEP elicited by other stimuli [22]. In this study, we reported abnormal pVEP in 33 of 63 patients (52.4%) with craniosynostosis, comparable to the prevalence of 60% of abnormal pVEP in craniosynostosis reported by Thompson et al. [11]. That higher prevalence may be attributed to the higher percentage of syndromic cases among the patient cohort (81 of 114; 71%), while syndromic/complex cases represented 35.8% of our cohort. This is reinforced by the finding of a higher percentage of abnormal pVEP in syndromic/complex cases (63.6%) than in patients with single suture, non-syndromic craniosynostosis (46.3%) in our study. Similar to the results obtained by Thompson et al. [11], no significant relationship between the pVEP grade and age was found in our patient cohort.

In our study, nine patients had evidence of intracranial hypertension, and of these, eight (88.9%) had abnormal pVEP, signifying the role of pVEP in detecting visual pathway dysfunction resulted from intracranial hypertension. This reinforces previous reports about the ability of pVEP to detect early visual dysfunction in craniosynostosis patients with raised ICP [13,18].

Surgical intervention for craniosynostosis can reverse visual pathway dysfunction through eliminating the compromising effect of increased intracranial pressure and cerebral hypoperfusion on the visual pathway before irreversible damage ensues. In our study, the role of surgery was obvious in reversing pVEP abnormalities detected preoperatively, with significant improvements in the pVEP amplitude and latency for the whole cohort. Interestingly, this improvement was not significant for patients with normal preoperative pVEP amplitude and latency. On the other hand, the postoperative decrease in latency was significant for patients with abnormally broad waveforms or delayed latencies, and the postoperative increase in amplitude was significant for patients with an abnormally low amplitude. This signifies the role of surgical intervention in reversing these visual pathway dysfunctions in these patients.

In our cohort, preoperative low amplitude was observed in 13 patients (grades 3 and 5), and of these, eight had syndromic/complex craniosynostosis. Reduced amplitude has been linked to hypoxia, resulting in a decreased number of functioning neurons along the visual pathway, as seen in anterior ischemic optic neuropathy [23,24]. In craniosynostosis, hypoxia can result from raised a ICP, resulting in intra-axonal edema with secondary optic ischemia [25,26]. Additionally, cerebral hypoperfusion may ensue in cases of syndromic craniosynostosis, due to the presence of OSA [27]. Low pVEP amplitude has been reported to increase, following adenotonsillectomy for the treatment of OSA in craniosynostosis [28]. 

Postoperatively, eight of nine patients with grade 3 and all patients with grade 5 normalized their amplitude within 6 months following craniosynostosis surgery. The importance of regular follow up for visual function with pVEP, even after surgery, can be clearly seen in serial pVEP testing performed for one of our patients, who showed persistent postoperative abnormal amplitude. This patient had Apert syndrome, and although he developed an initial increase in amplitude postoperatively, his subsequent pVEP tests showed a persistently low amplitude; a sleep study demonstrated OSA and the patient underwent several procedures for the dilatation of nasal and choanal stenoses.

Prolonged pVEP latency was demonstrated in 14 patients (grades 4 and 5) before surgery. Prolonged latency has shown to be the most common VEP abnormality in patients with hydrocephalus and idiopathic intracranial hypertension [29,30,31]. It can be explained by the slowing of rapid axoplasmic transport along the visual pathway, because of the raised ICP, as was described in experimental papilledema [32,33], which is analogous to conduction delay along the optic nerve observed in demyelinating disorders [24,34]. In mild cases, this interference with axoplasmic transport will result in a broadening of the pVEP waveform, or grade 2, seen in 10 patients of this cohort. A normalization of latency was found after shunt surgery in patients with hydrocephalus [29,35]. Similarly, we demonstrated the normalization of pVEP in all 10 patients with grade 2, and the normalization of latency was found in 13 of 14 patients with grades 4 and 5 postoperatively. The remaining patient showed persistent prolonged pVEP latency over a period of 18 months’ follow up. This persistence may be due to defective remyelination of demyelinated nerve fibers resulting from chronic compression [36]. This may indicate the importance of early intervention to reverse these abnormalities. Using a rat model with induced hydrocephalus, Del Bigio and Bruni showed that compensatory myelination was possible if treatment was instituted before the development of axonal injury [37].

It was to be noted that postoperative normalization of pVEP latency may be more gradual than the normalization of pVEP amplitude. Of the 13 patients with grades 4 and 5 who developed normal latency postoperatively, seven (53.8%) normalized their latency within 6 months, while 6 patients, although showing a decrease in latency within 6 months, developed normal latency over a period of 18 months after surgery. On the other hand, normal amplitude was achieved within 6 months for 12 of 13 patients (92.3%) with grades 3 and 5. Liasis et al. have demonstrated pVEP amplitude normalization within 6 months after intervention in eight patients with syndromic craniosynostosis [13]. 

The retrospective nature represents a possible limitation of this study. Additionally, we could not perform pVEP testing on a large number of healthy controls, as administering pVEP to healthy age-matched controls is time-prohibitive, given the significant resources that pVEP testing/analysis takes in this age population, in addition to the ethical issues.

## 5. Conclusions

In conclusion, our study showed that pVEP is an objective test for the early detection of the visual pathway dysfunction in patients with craniosynostosis, especially in the absence of clinical evidence of visual dysfunction in young children. Additionally, pVEP is a useful tool to monitor the efficacy of surgical treatment during the reversal of preoperative visual pathway dysfunction. The postoperative pVEP improvement includes waveform, amplitude, and latency, depending the on the preoperative abnormality, although normalization of the amplitude often occurs earlier than that latency, resulting in more gradual normalization of grades with abnormal latency.

## Figures and Tables

**Figure 1 jcm-08-01555-f001:**
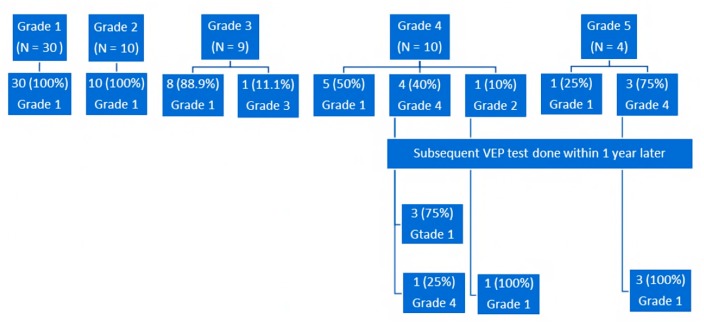
Postoperative changes in pVEP grades within 6 months for the whole cohort. Subsequent pVEP testing was performed within 1 year later for patients with grades 4 and 5 who had a persistent pVEP abnormality; pVEP, patterned, visually-evoked potentials.

**Figure 2 jcm-08-01555-f002:**
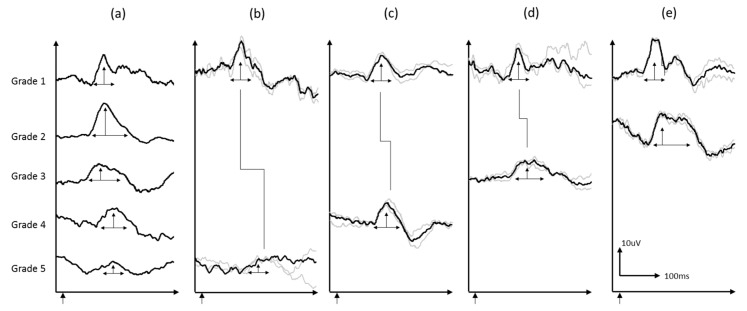
(**a**) Sample VEP waveforms grade 1–5, (**b**–**d**) are sample waveforms from individual subjects at baseline and follow-up: (**b**) grade 5 to 1, (**c**) grade 4 to 1, (**d**) grade 3 to 1, and (**e**) grade 2 to 1. Grey waves are individual trials and black waveforms are the grand average of the two reproducible trials.

**Figure 3 jcm-08-01555-f003:**
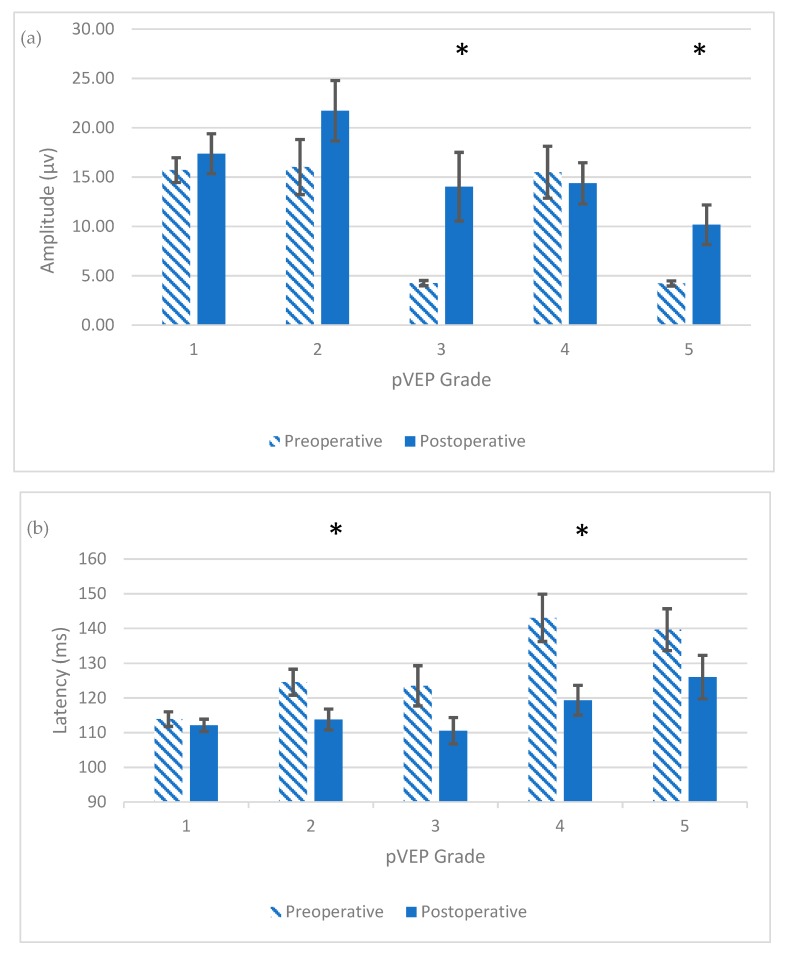
Postoperative changes in the means of pVEP amplitude (**a**) and latency (**b**), in each preoperative pVEP grade within 6 months after surgery, with the standard error of mean bars shown; (**a**) *, Significant postoperative increase in pVEP amplitude; *p* < 0.05; (**b**) *, Significant postoperative decrease in pVEP latency; *p* < 0.05.

**Table 1 jcm-08-01555-t001:** The grading of patterned, visually-evoked potentials (pVEP) according to the Thompson and Nischal classification [11].

Grade	Criteria	Number of Patients (%)	Mean Age at Preoperative VEP in Months (Range) *	Check Size Used
Grade 1	Normal VEP test	30 (47.6)	17 (4.4–82.1)	12.5′ (18 cases) 25′ (4 cases) 50′ (8 cases)
Grade 2	Normal amplitude and latency, but broad waveform	10 (15.9)	16.3 (2.8–75.5)	12.5′ (7 cases) 25′ (2 cases) 50′ (one case)
Grade 3	Reduced amplitude, broad waveform with normal latency	9 (14.3)	20.1 (2–121)	12.5′ (5 cases) 50′ (4 cases)
Grade 4	Normal amplitude, prolonged latency	10 (15.9)	16.1 (2–66.1)	12.5′ (4 cases) 25′ (2 cases) 50′ (4 cases)
Grade 5	Reduced amplitude, prolonged latency	4 (6.3)	19.8 (4.3–60.3)	12.5′ (2 cases) 50′ (2 cases)

* No significant relationship between age and preoperative pVEP grade, *p* = 0.51; Kruskal–Wallis test; VEP, visual evoked potentials.

**Table 2 jcm-08-01555-t002:** Pre- to post-operative changes in pVEP amplitude for the total cohort and different grades, within 6 months after surgery. Increased amplitude indicates improvement.

Grade	Total Cohort		Grade 1		Grade 2		Grade 3		Grade 4		Grade 5	
	Pre	Post	Pre	Post	Pre	Post	Pre	Post	Pre	Post	Pre	Post
Mean Amplitude (µv)	13.3	16.6	15.7	17.4	16.0	21.7	4.2	14.0	15.5	14.4	4.2	10.2
SD	±8.0	±10.0	±6.9	±11.1	±8.8	±9.7	±0.8	±10.4	±8.3	±6.6	±0.5	±4.0
*p* value *	*0.02*		0.24		0.11		*0.01*		0.37		*0.03*	

* Paired *t*-test; Standard deviation (SD).

**Table 3 jcm-08-01555-t003:** Pre- to post-operative changes in pVEP latency for total cohort and different grades within 6 months after surgery. Decreased latency indicates improvement.

Grade	Total Cohort		Grade 1		Grade 2		Grade 3		Grade 4		Grade 5	
	Pre	Post	Pre	Post	Pre	Post	Pre	Post	Pre	Post	Pre	Post
Mean Latency (ms)	123.2	114.2	113.9	112.1	124.5	113.8	123.5	110.6	143.0	119.3	139.6	126.0
SD	±17.7	±11.1	±11.5	±9.7	±11.9	±9.5	±17.4	±11.4	±21.6	±13.6	±12.0	±12.5
*p* value *	<0.001		0.18		0.02		0.03		0.007		0.07	

* Paired *t*-test.

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
