# Peer review of "Serial, Visually-Evoked Potentials for the Assessment of Visual Function in Patients with Craniosynostosis"

_jcm, 2019, doi:10.3390/jcm8101555_

Round 1
Reviewer 1 Report
In general, the authors addressed my concerns adequately. There are still a few points to be made, given below:
The authors explain in the response letter how reproducibility was determined, but this explanation along with the appropriate reference should be included in the manuscript. Also, the explanation for the measurement of broadening in the waveform is given in the response letter along with the reference, but it would be helpful to have that information in the current manuscript. The authors provided the counts of each check size used to elicit each type of response in the response letter, but this would be useful information to include in the manuscript as well. Although the authors state “we feel that parametric testing is appropriate,” they do not explain the statistical reasoning for drawing that conclusion – was it based on examination of frequency histograms? Although the authors state that the “typographical errors have been corrected,” the words “raised” and “examination” on line 135 are still misspelled. The authors now include error bars in Figure 2, but they should specify if those bars represent standard errors of the mean (i.e., ± 1 SE).
Author Response
Reviewer 1
Comments and Suggestions for Authors
In general, the authors addressed my concerns adequately. There are still a few points to be made, given below:
The authors explain in the response letter how reproducibility was determined, but this explanation along with the appropriate reference should be included in the manuscript.Reply: This has been added (on line 87) along with the appropriate reference.
Also, the explanation for the measurement of broadening in the waveform is given in the response letter along with the reference, but it would be helpful to have that information in the current manuscript.Reply: This has been added in the text on line 102, with the reference has been added at the end of the paragraph, that is the source of this classification.
The authors provided the counts of each check size used to elicit each type of response in the response letter, but this would be useful information to include in the manuscript as well.Reply: This has been added in the text on lines 92-95 and Table 1.
Although the authors state “we feel that parametric testing is appropriate,” they do not explain the statistical reasoning for drawing that conclusion – was it based on examination of frequency histograms?Reply: The basis of using the parametric tests was based upon frequency histograms that as a group were similar for amplitude and latency before and after surgery.
Although the authors state that the “typographical errors have been corrected,” the words “raised” and “examination” on line 135 are still misspelled.Reply: These have been corrected.
The authors now include error bars in Figure 2, but they should specify if those bars represent standard errors of the mean (i.e., ± 1 SE).Reply: The graphs have been reformatted to show the standard error of mean bars. This has been also specified in the legend of the figure.
Reviewer 2 Report
The authors addressed all issues raised by the reviewers. One minor suggestion would be that the authors also state in the discussion why no control group was used, since this is easy to justify. Otherwise, I have no further comments.
Author Response
Reviewer 2
Comments and Suggestions for Authors
The authors addressed all issues raised by the reviewers. One minor suggestion would be that the authors also state in the discussion why no control group was used, since this is easy to justify. Otherwise, I have no further comments.Reply: This has been added.
This manuscript is a resubmission of an earlier submission. The following is a list of the peer review reports and author responses from that submission.
Round 1
Reviewer 1 Report
The authors investigated the effect of surgery for craniosynostosis on visual evoked potentials. Please find my comments below:
In Table 1, the age range for each group should be provided.
page 4, line 124, the number of patients that had grade 5 is missing.
Although the conclusions are supported by the results, I would suggest to also mention the limitations of the study and discuss them. This would be on the one hand the retrospective character and on the other hand the lack of a control group, which is of course for obvious ethical reasons not possible to include.
Author Response
Reveiwer 1
Comments and Suggestions for Authors
The authors investigated the effect of surgery for craniosynostosis on visual evoked potentials. Please find my comments below:
Reply: This has been added.
page 4, line 124, the number of patients that had grade 5 is missing.Reply: This has been added.
Although the conclusions are supported by the results, I would suggest to also mention the limitations of the study and discuss them. This would be on the one hand the retrospective character and on the other hand the lack of a control group, which is of course for obvious ethical reasons not possible to include.Reply: A sentence about limitations has been added before the conclusion.
Reviewer 2 Report
This manuscript demonstrates the value of recording transient visual evoked potentials to contrast-reversal of a patterned (checkerboard) stimulus (pVEP) for the assessment and monitoring of individuals with craniosynostosis. The authors report on pVEPs obtained from patients as young as two-months old before and following craniofacial surgery. Results showed that the majority of these patients exhibit abnormal pVEPs prior to surgery, and that surgical intervention yielded positive effects on pVEP measures of amplitude, latency, and waveform broadening in follow-up testing within a six-month period for most of the impaired cases. Previous work has demonstrated that visual dysfunction can be detected using the pVEP even in cases of craniosynostosis with normal optic disc appearance and visual acuity. Early assessment with a sensitive functional test such as this one holds the promise of preventing permanent neural damage, and the current work shows that deficits in visual function can be reversed and function normalized with effective treatment.
The approach taken in this study is to apply conventional time-domain analysis to the pVEP. That is, a few time points are selected to characterize the response in terms of peak times and amplitudes. An experienced electrophysiologist needs to visually inspect the waveform to ensure choosing appropriate points to measure. A waveform contaminated by electrical noise can complicate the decision process and add considerable variability to the measurements. Thus, this ‘objective’ test has a major subjective component to it. This manuscript requires additional details to clarify the methods applied to strengthen the authors’ claims, and the following points should be addressed:
Was line noise filtered out of the recordings, and if so, by what method? What was the gain of the amplifiers? What was the model of the Espion equipment used to collect data? The authors state that “A minimum of two reproducible trials were recorded for each stimulus.” How was reproducibility determined? Was it simply the subjective judgment of the experimenter following visual inspection? How was broadening of the waveform measured? What criteria were used to select start and end points in the time-domain record? Initially, different check sizes were used as stimuli and the ‘threshold check size’ for an individual was used for the remainder of testing. How was that smallest check size that yielded an ‘observable’ response chosen? Given that the pVEPs were graded into five categories, what was the count of each check size used to elicit each type of response? Presumably, inclusion of pVEPs to different check sizes increased the variability of measurements in response waveforms – e.g., larger checks typically elicit shorter peak times than do small checks. It would be useful to include example waveforms for each category of response to illustrate the quality of the recordings and the criteria used for measurement. Pre- and posttreatment responses for representative individuals would be informative. On line 94, “Table 1” from Thompson et al. is mentioned, but it appears the authors meant Figure 2 from that article. In the Statistical Analysis section, it is stated that both nonparametric (Kruskal-Wallis) and parametric (paired t-test) statistics were applied, but did the frequency distributions of response measures support the use of the parametric statistic or should a nonparametric test have been used there as well. Information included in the tables is repeated in the text, which is unnecessary. There are typographical errors in lines 124, 127, 129, and 132. In Figure 2, do the bars indicate mean values? If so, error bars or confidence intervals should be added. If they represent median values, interquartile ranges should be added or change the graphs to boxplots.The pVEP is a powerful tool for the assessment of brain function on the time scale of milliseconds. For future work, the authors might want to consider the inclusion of frequency-domain measures that can quantify the pVEP response in a more objective way. Magnitude of response can be characterized by the power in a range of harmonic frequency components extracted by means of Fourier analysis. Phase values of these components can be used to calculate estimates of time delay. These frequency-domain measures are computed by the Fourier transform without subjective input from the electrophysiologist and capture much of the informational content of the response, in contrast to the limited information obtained by taking a few measurements at select time points.